# Hyponatremia in Infectious Diseases—A Literature Review

**DOI:** 10.3390/ijerph17155320

**Published:** 2020-07-23

**Authors:** Anna L. Królicka, Adrianna Kruczkowska, Magdalena Krajewska, Mariusz A. Kusztal

**Affiliations:** 1Faculty of Medicine, Wroclaw Medical University, 50-367 Wroclaw, Poland; anna.krolicka01@gmail.com; 2Department of Nephrology and Transplantation Medicine, Wroclaw Medical University, 50-367 Wroclaw, Poland; magdalena.krajewska@umed.wroc.pl (M.K.); mariusz.kusztal@umed.wroc.pl (M.A.K.)

**Keywords:** hyponatremia, infections, infectious diseases, syndrome of inappropriate antidiuretic hormone secretion

## Abstract

Hyponatremia is one of the most common water–electrolyte imbalances in the human organism. A serum sodium concentration threshold of less than 135 mmol/L is diagnostic for hyponatremia. The disorder is usually secondary to various diseases, including infections. Our review aims to summarize the diagnostic value and impact of hyponatremia on the prognosis, length of the hospitalization, and mortality among patients with active infection. The scientific literature regarding hyponatremia was reviewed using PubMed, ClinicalKey, and Web of Science databases. Studies published between 2011 and 2020 were screened and eligible studies were selected according to the PRISMA (Preferred Reporting Items for Systematic Reviews and Meta-Analyses) statement and specific inclusion criteria. The most common infections that were associated with hyponatremia were viral and bacterial infections, including COVID-19 (coronavirus disease 2019). The etiology varied according to the infection site, setting and patient cohort it concerned. In several studies, hyponatremia was associated with prolonged hospitalization, worse outcomes, and higher mortality rates. Hyponatremia can also play a diagnostic role in differentiating pathogens that cause a certain infection type, as it was observed in community-acquired pneumonia. Although many mechanisms leading to hyponatremia have already been described, it is impossible with any certainty to ascribe the etiology of hyponatremia to any of them.

## 1. Introduction

Hyponatremia is defined as a low sodium concentration in blood serum. The sodium concentration below the lower reference limit of 135 mmol/L is diagnostic for this disorder. Hyponatremia is one of the most commonly known water–electrolyte imbalances in the human organism. Its prevalence reaches 15–30% among hospitalized patients, which is even higher in the intensive care units (ICU), where it reaches up to 40% [1,2]. The incidence of hyponatremia in the general population is estimated at about 1.74%, according to National Health and Nutrition Examination Survey [3]. It is worth mentioning that in both in-patient and out-patient settings, hyponatremia is an acknowledged mortality predictor, although its causes are different in these populations [4]. Furthermore, hyponatremia prevalence was shown to be positively associated with comorbidity, expressed as Charlston Comorbidity Index, and was more profound among the most moribund patients [5]. Winzler et al. also emphasize that 1-year mortality among patients admitted to hospital due to severe hyponatremia (<125 mmol/L) reached 20% in their study, which underlines the gravity of this pathology and the need for adequate causal treatment. A recently performed meta-analysis has confirmed that the improvement of hyponatremia is independently associated with the reduction of all-cause-mortality risk [4].

Hyponatremia may occur as a result of numerous disorders [6]. These include common infectious diseases as well as endocrine, nutritional, metabolic, cardiovascular, renal or hepatic pathologies. It is worth mentioning that hospital-associated hyponatremia includes admission hyponatremia and hospital-acquired hyponatremia, which may be associated with different outcomes [7]. 

Patients presenting with hyponatremia may often be asymptomatic, which applies especially to elderly patients or patients with mild hyponatremia. On the contrary, most moderate to severe, as well as acute cases of hyponatremia present a broad spectrum of symptoms as nausea, confusion, headache, vomiting, which in severe cases may progress to delirium, impaired consciousness, and seizures [8]. Due to its non-specific presentation, many healthcare professionals are not aware of the association of hyponatremia and ongoing inflammation [9].

The primary purpose of this review was to perform a meta-analysis, which would elucidate the impact of hyponatremia on the prognosis, mortality, and outcomes among in-patients with an acute infective process. However, after selecting eligible papers and analyzing the available data, these were deemed insufficient for a meta-analysis by a certified statistician and instead a review has been written instead. Although the paper does not strictly meet the definition of a systemic review, we have decided to still adhere to the PRISMA (Preferred Reporting Items for Systematic Reviews and Meta-Analyses) model as to ensure the highest quality of the selected material and scientific content.

### 1.1. Pathogenesis of Hyponatremia

Sodium and potassium are the main osmotically active inorganic substances and their plasma concentrations are maintained within a narrow range [6,10]. The water transfers between intra and extracellular compartments are driven by an osmotic pressure gradient exerted by osmotically active solutes, including sodium. The amount of active osmolytes in serum is expressed by tonicity, which is a substantial parameter utilized for distinguishing between hyponatremia causes. Thus, for diagnostic purposes hyponatremia can be divided into hypotonic and non-hypotonic hyponatremia, as presented on Figure 1 [1,6].

### 1.2. Hypotonic Hyponatremia

Hypotonic hyponatremia is associated with serum osmolality values of less than 275 mOsm/kg [1]. It can be classified into the following three groups: hypovolemic, euvolemic, or hypervolemic hypotonic hyponatremia. The diagnosis is made depending on extracellular volume status. The most commonly utilized tools for volemic estimation are clinical assessment, inferior vena cava or central venous pressure measurement, as well as a lung ultrasound [11]. Each of these methods has certain limitations and should be used supplementary to each other to ensure proper volemia assessment [6]. 

#### 1.2.1. Hypotonic Hypovolemic Hyponatremia

This type of hyponatremia is characterized by extracellular fluid volume depletion secondary to a deficit of both total body sodium and water. Baroregulated secretion of vasopressin in response to hypovolemia and inappropriate oral or parenteral hypotonic fluid intake leads to water retention, causing hyponatremia [12]. Solute loss can be classified as renal, due to salt-wasting nephropathy or mineralocorticoid deficiency, and non-renal causes. Exemplary causes of non-renal hypovolemic hyponatremia include skin losses as a consequence of burns or perspiration and gastrointestinal losses such as diarrhea or vomiting, which are often associated with infections [1,6]. Cerebral salt wasting syndrome (CSWS) is also a cause of hypovolemic hyponatremia. CSWS is sporadically observed in patients with intracranial pathologies. A variety of infections of the central nervous system (CNS) such as tuberculous meningitis, poliomyelitis, and toxoplasmosis have also been linked to CSWS [13]. The clinical signs of this type of hyponatremia include dry mucous membranes, decreased skin turgor, tachycardia, hypotension, raised urea, and creatinine levels in serum [12]. 

#### 1.2.2. Hypotonic Euvolemic Hyponatremia

Euvolemic hyponatremia is the most common cause of hyponatremia among hospitalized patients [12]. The most frequent underlying disorder is Syndrome of Inappropriate Antidiuretic Hormone Secretion (SIADH). SIADH is defined as the presence of hypoosmolality when urine osmolality is inappropriately high and there is no evidence of renal salt wasting or hypovolemia in an individual with normal organ functions. In SIADH, high antidiuretic hormone (ADH) activity is considered to be inappropriate, since there is no identifiable osmotic or hemodynamic stimulus for its secretion [6]. The most common causes of SIADH are malignancies, pulmonary disorders such as community-acquired pneumonia, central nervous system disorders, and drugs [6,13,14]. In HIV/AIDS patients, opportunistic infections of the pulmonary tract or CNS can also induce the release of excessive ADH [15].

In rare cases, patients may develop hyponatremia due to true volume depletion, glucocorticoid deficiency, polydipsia, malnutrition, or severe hypothyroidism [6,16,17]. Sometimes, long-lasting physical activity and diet poor in sodium may also be a reason of hypovolemic or euvolemic hyponatremia.

#### 1.2.3. Hypotonic Hypervolemic Hyponatremia

Hypervolemic hyponatremia occurs in states such as decompensated heart failure, advanced liver disease or renal disease, where excessive water causes plasma dilution [8]. In liver diseases with cirrhosis at their end-stage, lacking mineralocorticoid elimination leads to excessive water and salt retention, while low blood albumin levels induce a water shift from vessel lumen into the tissue compartment [18]. Such a constellation leads to hypervolemia, edemas, and finally hypervolemic hyponatremia. On the contrary, in heart failure, decreased cardiac output and venous congestion are the primary underlying pathologies, leading to impaired water elimination in the kidneys, volume overload, and secondary sodium dilution [19]. Thus, the treatment of this hyponatremia type relies mainly on diuretics use and proper management of the underlying disorder. 

### 1.3. Non-Hypotonic Hyponatremia

There are two types of nonhypotonic hyponatremia: pseudohyponatremia and hypertonic hyponatremia. Hypertonic hyponatremia occurs due to an increased concentration of osmotic active agents as mannitol or excess glucose. The mentioned sugars earlier are osmotically active substances and they do not cross freely into the cell, causing water movement [20]. As a consequence, water transfers from the intracellular compartment to the extracellular compartment.

Pseudohyponatremia is hyponatremia occurring along with isotonic osmolality. It is mainly caused by the displacement of serum water because of elevated concentrations of serum lipids or proteins. Expansion of the lipid or protein portion decreases the water portion resulting in artificially lowered serum sodium compared with serum measured directly in the water portion of plasma volume [6,20]. 

## 2. Materials and Methods

We performed a systematic search of the literature in June 2020. Research terms comprised a combination of words: “hyponatremia”, “adults”, “infection”, and “SIADH”. We analyzed the articles, which were published between 2011 and 2020, as a similar meta-analysis was carried out and released in 2011 [9]. Only English papers were reviewed. The articles were evaluated according to the Preferred Reporting Items for Systematic Reviews and Meta-Analyses (PRISMA) protocol (Figure 2) and were included after fulfilling following inclusion criteria.

Inclusion criteria:Age >18 years old,Infection and concomitant hyponatremia in the study group,Original articles.

Exclusion criteria:Case reports,Cases series.

Finally, 62 articles were found eligible as presented on the flowchart below and representative information was investigated.

## 3. Hyponatremia Due to Infections 

During the literature search, we identified the following infection-related causes of hyponatremia: viral infections, bacterial infections, fungal infections, and protozoal infections. The most prevalent microorganisms within each group are presented in Table 1. In several papers, hyponatremia has been reported as a concomitant finding in patients with mixed infections.

### 3.1. Viral Infections

#### 3.1.1. Influenza Virus and other Respiratory Viruses

In general population community-acquired pneumonia (CAP) is one of the most common viral infections associated with hyponatremia. Although CAP is mainly caused by bacteria such as *Streptococcus pneumoniae* or *Legionella pneumophila,* viruses still remain a relevant etiologic factor for CAP and secondary bacterial pneumonias. The main viral pathogens causing CAP are rhinovirus, respiratory syncytial virus, influenza viruses, parainfluenza viruses, and adenovirus [21]. Yet, in many cases the cause of CAP remains unidentified. CAP was one of the complications of influenza A(H1N1) infection in 2009 during the pandemic and it was associated with high morbidity, mortality, and frequent admission to ICU [22]. According to data, most of the incidences of influenza cases during the post-pandemic season period occurred in the first weeks of 2011 with dominant viral CAP etiology [22]. Previous studies revealed that among patients diagnosed with CAP, an etiological diagnosis could be established only in 42.2% of them, with 21.9% of them being infected by bacteria, 16.7% by viruses, and 4.8% of patients had mixed infection (bacterial and viral). Patients that required ICU admission were mainly affected by influenza A (H1N1) virus and influenza B virus infection. What is more, the patient cohort with viral pneumonia was younger and was less moribund than the group with non-viral pneumonia. Those with viral and mixed pneumonia, not only required ICU admission more often but also had a higher in-hospital mortality. Severe disease occurred in 16.6% of patients. Hyponatremia was independently associated with disease severity and was found in 17.6% of patients with viral pneumonia, 13.8% of patients with bacterial pneumonia, 11.1% of patients with mixed pneumonia, and 5.4% of patients with pneumonia of undetermined etiology. Hyponatremia at admission was also associated with a longer hospital stay [23]. According to these findings, hyponatremia occurs most frequently when patients suffer from viral and bacterial pneumonia and is less frequent in patients with unknown etiology of CAP [22].

Low serum sodium concentrations along with other dyselectrolytemias were also identified in patients with avian-origin influenza A(H7N9). On initial presentation, hypokalemia, hyponatremia, and hypocalcemia were common. Almost half of the patients with confirmed H7N9 infection (43.6%) were hyponatremic [24].

#### 3.1.2. HIV Infections

Hyponatremia is a very common electrolyte abnormality among patients with AIDS [15]. According to literature, hyponatremia may occur in 20–80% of hospitalized patients with HIV disease or AIDS [15]. A common comorbidity in HIV-infected patients are pulmonary or CNS infections, which may result in SIADH. It may explain why hyponatremia is inherent in this disease.

Hyponatremia also occurs as a result of adrenal insufficiency, renal disorders, and gastrointestinal sodium loss due to diarrhea and vomiting, which are observed frequently in AIDS patients [25,26,27]. According to studies, there is a positive correlation between serum sodium concentrations and the number of CD4+ cells, as hyponatremic patients presented with lower CD4+ cell count and showed a negative correlation with the WHO clinical stage [27,28]. This may suggest that serum sodium concentrations can be used as an indicator of the progression of HIV-infection or AIDS, although it is not a specific marker. What is more, a significantly higher proportion of patients with hyponatremia are hospitalized at first contact, the time between the diagnosis and their first hospitalization is shorter, and they have a higher incident hospitalization rate compared to normonatremic patients [28].

Despite numerous observations that link hyponatremia development with HIV-disease progression, it still remains a topic of controversy whether hyponatremia does directly increase the mortality among HIV infected patients. Currently, there is no strong data that would confidently support such a thesis. Braconnier et al. reported a similar amount of HIV-related deaths in hypo- and normonatremic patients at baseline [28]. In the same study it has also been reported that hyponatremic patients presented a higher all-cause mortality, which has been linked to lower CD4 count and higher prevalence of hepatitis C. On the other hand, moderate to severe hyponatremia was shown to be associated with higher long-term mortality [27]. What is more, the majority of HIV-infected deaths occurred within the first 3 months of anti-retroviral therapy initiation, and hyponatremia was independently associated with mortality. It has also been observed that serum chloride and serum sodium levels are regulated together, and hyponatremia is frequently associated with hypochloremia, which leads to higher mortality, compared to patients with only one electrolyte abnormality. Dao et al. have also suggested that subsequent mortality may also be caused by undiagnosed conditions primary to these electrolyte abnormalities and not by the electrolyte abnormality itself [29].

Despite the fact that SIADH is considered to be one of the factors leading to hyponatremia in HIV-infections or AIDS, no differences between the ADH, renin, or aldosterone levels of HIV patients and those of non-HIV-infected subjects were observed [30]. Currently, there is no strong data that would support the thesis that hyponatremia may contribute directly to increased mortality among HIV-infected patients. That is why it should only be treated as a surrogate marker of the severity of the underlying disease.

#### 3.1.3. Coronaviruses Infections

Early reports analyzing COVID-19 (Coronavirus Disease 2019) show that hyponatremia may be a common laboratory finding among infected patients [31]. However, different reports show that hyponatremia in this patient cohort may not only be associated with COVID-related pneumonia, but may also be attributed to inadequate dietary intake by COVID-19 patients or the gastrointestinal presentation of the disease. In patients described by Yiqun Wu et al., 17.6% of the study group presented with hyponatremia. Irrespectively from the hyponatremia etiology, it appeared more frequently in patients with a hospital stay longer than 14 days. Nonetheless, in a multivariate analysis, it was not recognized as an independent risk factor associated with long-term hospitalization in patients with COVID-19. [32]. Berni et al. reported that among non-ICU, COVID-patients, hyponatremia correlated positively with oxygenation index (PaO2/FiO2 ratio) and inversely with IL-6 levels. The investigated patient cohort in this study was partially qualified for tocilizumab (anti-IL6 antibody) treatment, based on IL-6 levels. Interestingly, the initiation of such treatment was associated with significant improvement of hyponatremia at 48 h, as compared to control group [33]. The authors of this study also emphasize that hyponatremic patients have presented worse outcome (more ICU-transfers and more non-invasive ventilation initiations) than their normonatremic counterparts. No information about long-term outcomes in patients treated with tocilizumab have been presented in this study.

Therefore, further research has to be carried out to state the role of hyponatremia in this new disease and to explore its possible association with response to treatment, morbidity, time of hospitalization, and mortality.

#### 3.1.4. Other Viral Infections

Other viral infections associated with hyponatremia include human herpesvirus-6 (HHV-6), SFTS virus (SFTSV), hantavirus, herpes simplex virus (HSV), and Ebola virus infections [34,35,36,37,38,39,40]. 

Severe fever with thrombocytopenia syndrome (SFTS) is an infectious disease caused by the SFTSV. Retrospectively, SFTSV infections are uncommon but are associated with high mortality, reported at the level of 33% [34]. Hyponatremia was found to be an independent risk factor of death in SFTS patients. In the non-survivor group, hyponatremia occurred in 12 cases (60%), and in the survivor group in eight cases (20%). Generally, electrolyte imbalances including hyponatremia are common in SFTS and are related to higher disease severity and worse outcome.

Many papers also discuss acute CNS infections, with the main focus on encephalitis. The most common viral pathogens responsible for CNS infections are HSV, Enterovirus, and Varicella zoster virus (VZV) [35]. Hyponatremia, both hypovolemic and euvolemic is associated with all viral CNS infections due to vomiting, SIADH, or fever. That is why CNS infections are related with high incidence of hyponatremia that reaches 30–66% [36]. In general, hyponatremia is associated with CNS infections in about 30–66% of cases [36]. It has been observed that low serum sodium levels appear more frequently in CNS HSV infections. Thus, it has been suggested that serum sodium could be used to differentiate the HSV vs. non-HSV encephalitis etiology [36,37].

The reactivation of human herpesvirus-6 latent infection has been reported after allogeneic hematopoietic stem cell transplantation, leading to CNS disorders such as myelitis and encephalitis [38]. At the onset of HHV-6 encephalitis, serum sodium level decreased in all patients with encephalitis (median of 129.1 mEq/L) and it was lower than in patients with myelitis. Some factors, for instance antineoplastic agents, glucocorticoids, calcineurin inhibitors, diuretics, and malnutrition can possibly contribute to hyponatremia. What is more, SIADH occurs in a variety of CNS disorders or infections and it may be involved in the development of hyponatremia associated with HHV-6 encephalitis [38]. Some scientists propose that hyponatremia could be a sign of HHV-6 encephalitis, but not of myelitis after allogeneic hematopoietic stem cell transplantation.

Hantavirus causes hemorrhagic fever with renal syndrome. Severe hyponatremia has been reported to be associated with this disease, predicting the development of acute kidney injury (AKI) [39].

The lowered concentration of sodium, which is a common laboratory abnormality in Ebola patients, is most probably associated with sodium and water loss due to severe vomiting and diarrhea [40]. Retrospective studies have shown that hyponatremia was present in about 44% of patients diagnosed with Ebola virus infection on admission. In the course of hospitalization its prevalence has increased up to 78% [40,41,42]. All of the patients received treatment during hospitalization. 

### 3.2. Fungal Infections

Only one article was found, which described hyponatremia in a fungal infection—cryptococcosis. Cryptococcal infection has two most common manifestations: cryptococcal meningitis or pulmonary cryptococcosis. As a result, many mechanisms leading to hyponatremia development in this patient group must be considered. As multivariable analysis showed, hyponatremia at presentation is statistically significantly associated with cryptococcal infection [43].

### 3.3. Overlapping Infection

Hyponatremia has long been recognized as a complication of malaria [44]. Both malaria and HIV infections independently affect the electrolyte profiles, but not much is known how these two infections together affect the serum sodium level. According to studies, comparing four groups of patients, HIV-uninfected with uncomplicated malaria, HIV-uninfected with severe malaria, HIV-infected with uncomplicated malaria, and HIV-infected with severe malaria, there is no statistically valuable connection between HIV and malaria causing hyponatremia. Separately and together, serum sodium level is similar in these four groups [45].

Toxoplasmic encephalitis (TE) is the most frequent neurologic disorder in patients with advanced HIV infection [46]. Alterations in serum sodium regulation are a common feature in neurologic disorders like TE. Neurologic-associated hyponatremia could be secondary to SIADH. In one article, among the patients diagnosed with toxoplasmosis, hyponatremia on admission was verified in 46.7% and severe hyponatremia in 10.8% of the cases [47]. What is more, HIV-infected and TE populations are at risk of developing acute kidney injury (AKI) mainly because of the treatment for TE, where the first line medicaments are the sulfonamides. Hyponatremia was proven to be an important risk factor for developing AKI. It triggers the release of ADH, leading to increased urine concentration, which in turn predisposes to sulfadiazine crystal deposition in kidneys. In this study survivors differed from non-survivors in serum sodium levels with a significantly higher prevalence of hyponatremia (35.4% vs. 92.3%; *p* < 0.0001). A multivariable analysis showed that hyponatremia on admission was an independent risk factor for hospital mortality [47].

Tuberculosis is another common co-infection in HIV/AIDS patients, and it may be also associated with hyponatremia that occurred in 28.4% cases, which, together with active tuberculosis and WHO clinical stage 4 status, is thought to be an important risk factor for mortality within 6 months of admission. Additionally, it is suggested that hyponatremia was not the direct cause of death but an indicator of the critical status of the patient, as deaths still occurred even after the patient’s serum sodium levels had been corrected [48].

### 3.4. Bacterial Infections

The most common topic among articles concerning bacterial infections was CAP, where low serum sodium is one of the most frequent laboratory findings [49]. There are many theories about the association of hyponatremia with infections; however, the etiology remains unclear despite the many attempts to identify it. There is a study that ruled out SIADH being the cause of hyponatremia in Legionnaires’ disease by measuring and finding no connection between CT-ProVasopressin and sodium levels [50].

Some studies suggest that hyponatremia can be a useful marker in differentiating *Legionella pneumophila* from *Streptococcus pneumoniae* or other non-legionella pneumophila infections [51,52]. The utility of pneumonia pathogen identification relies on the fact that *L. pneumophila*, unlike other pneumonia-typical pathogens, seems to be strongly associated with severe respiratory symptoms as well as sepsis with multiorgan dysfunction [53]. First line antibiotics for pneumonia treatment, such as amoxicillin or other beta-lactams, do not possess antimicrobial activity against this pathogen. Thus, easy and commonly available tests that could predict *L. pneumophila* infections with good sensitivity could prove useful, as different treatment is required by this pathogen.

Patients with *L. pneumophila* tend to have lower serum sodium levels than patients with pneumonia caused by other pathogens [52]. It has also been shown that hyponatremia is positively associated with *L. pneumophila* Urinary Antigen Test positive result [54]. Still, there is no perfect scoring system that would differentiate these bacteria from others. In the six-point scoring system proposed in the cited article, the cut off value of serum sodium is 133 mmol/L, while the four-point scoring system defines it at 137 mmol/L. However, there are studies proving that the six-point scoring system is of no use and in many cases may lead to a misdiagnosis. That is why a microbiological investigation still remains the best way of confirming *Legionella pneumonia* infection [52,55,56].

Nonetheless, hyponatremia in CAP is associated with higher mortality and longer in-hospital stay [57]. Although in most cases hyponatremia was only mild to moderate, it still resulted in relevant clinical symptoms [49].

Other articles about hyponatremia in bacterial infections included peritonitis, tuberculosis, murine typhus, sepsis, bacterial meningitis, skin, and soft tissue infection, Guillain Barre Syndrome, Mediterranean spotted fever (MSF), and scrub typhus [58].

Low serum of sodium is also common in Spontaneous Bacterial Peritonitis (SBP). A retrospective case-control study revealed that, among cirrhotic patients, those with SBP suffered more frequently from hyponatremia than non-SBP patients [59]. Severe hyponatremia (<125 mmol/L) is also reported to be a risk factor of developing SBP, which may be related to higher volume overload with sodium dilution as well as intestine mucosa edema, allowing spontaneous bacterial migration through its wall [60]. Moreover, hyponatremia is commonly observed in 27% of patients who suffer from peritoneal-dialysis-related peritonitis (PDRP). In a retrospective analysis, serum sodium level was positively correlated with serum albumin and phosphate levels. In the hyponatremic group, the in-hospital stay was lengthened, and the mortality rate was higher [61,62]. Another retrospective study shows that baseline hyponatremia (<130 mmol/L) is independently associated with refractory peritonitis [63].

In miliary tuberculosis, hyponatremia is reported to be associated with an unfavorable clinical outcome and predicts the need for mechanical ventilation and critical care intervention [64]. While in miliary tuberculosis the reasons for hyponatremia remain unclear, there are studies which precisely define its etiology in tuberculous meningitis (TBM). About 45% of TBM patients had hyponatremia, with CWS and SIADH being its two main causes. This study states the predictors of hyponatremia in TBM, which are lower Glasgow Coma Scale, higher frequency of infarction in both anterior and posterior circulation, and higher incidence of basal exudates [65].

Another bacterial infection in which hyponatremia is observed is murine typhus. Its prevalence reaches up to 44% of patients, occurring more often in children than in adults [66,67].

Among patients suffering from Guillain–Barre Syndrome, the elderly (more than 60 years old) were characterized by a higher rate of hyponatremia compared to the younger group (18–59 years old) with a ratio of 25% vs. 10.2%, (*p* < 0.01) [68].

An analysis of the epidemiologic trend of adult bacterial meningitis reveals that the most common infections were monomicrobial, caused by Staphylococcal species. Hyponatremia was also a common laboratory finding in this disease [69].

It is well known that patients with liver cirrhosis have worse immunity and are prone to infections. These include skin and soft tissue infection, which appear to be associated not only with hyponatremia, but also renal failure. In most cases, the infection primary site is localized on the lower extremity. In patients with skin and soft tissue infection, hyponatremia occurs significantly more often, and sodium concentration, precisely the Model of End-Stage Liver Disease sodium, is associated with mortality among infected patients [70].

One article suggested that hyponatremia is of no prognostic value to predict the progression of sepsis to severe sepsis or septic shock [71].

Elderly patients with MSF suffered more frequently from hyponatremia than non-elderly MSF patients with a prevalence of 20.0% vs. 8.1% (*p* = 0.014), respectively [72].

### 3.5. Protozoal Infections

Malaria is a protozoal disease, in which hyponatremia is commonly observed in approximately 60% of the cases [73]. A hypothesis has been proposed that it is caused by a dysregulation of vasopressin release. It is also suggested that serum sodium of 131 mmol/L and lower characterizes severe falciparum malaria disease [74]. Yet, the WHO guidelines for diagnosing severe hyponatremia do not take into consideration sodium levels [75]. One of the studies identified hyponatremia as a mortality predictor, which predicted death at the cut off of 126 mEq/L with the specificity of 78.8% and sensitivity of 81.1% [73].

Another study identified hyponatremia in 66% of malaria contracted patients during antineoplastic therapy [76].

## 4. Other Potentially Infection-Related Causes of Hyponatremia

Many medical departments, as ICU or internal wards, deal with moribund patients who frequently develop both hyponatremia and infections in the course of hospitalization. Due to high prevalence of neoplasms, AKI, pulmonary pathologies, and other comorbidities, the etiology of hyponatremia often remains unclear. Thus, the knowledge about alternative hyponatremia causes and the distinction between them is crucial for proper decision making and patient management. 

### 4.1. SIADH

One retrospective study aimed to analyze the difference between hyponatremia in SIADH and hyponatremia in AKI. 24.9% of patients with SIADH were diagnosed with an infection, either acute or chronic. In general, patients with SIADH were characterized by a higher median survival time compared to those with AKI. Among patients who died within the first year, 23.8% had an infection. Furthermore, mortality in the first 2.5 years after diagnosis was higher among AKI patients, but after 2.5 years it became similar in both groups [77].

Another study strived to investigate the etiology of SIADH. The main identified causes of this disorder were cancer (27.7%), drugs (26.5%), idiopathic (15.9%), pulmonary infections (12.3%), pain and nausea (10.4%), and CNS disorders (7.2%). All these groups were comprehensively described. As far as pulmonary infections were concerned, these patients presented the highest median of serum sodium concentration, most of them (45.5%) suffered from moderate hyponatremia, and their long-term (30 months) survival-rate was extremely low with 36.8%. According to this study, the etiology of SIADH is an important prognostic factor of survival [78].

The mentioned SIADHs etiology is of great importance since it determines the patient’s prognosis. That is why one center retrospective study, focused on stating the characteristics of patients diagnosed with idiopathic SIADH was performed, in an attempt to re-diagnose those patients. The researchers managed to establish a new diagnosis only in 11 out of 99 patients, while in 88 out of 99 the etiology remained idiopathic. In the group of re-diagnosed patients, seven had a malignancy detected, two had chronic pulmonary infections, one had benign brain space-occupying lesion, and one had Kikuchi syndrome [79].

Although SIADH is known to be associated with certain infections and presents a well-known clinical range of symptoms, its precise mechanism still remains uncertain. Studies designed to present a potential correlation between ADH-related peptides (as CT-ProVasopressin) and sodium serum levels have failed to do so. Thus, the pathophysiology of SIADH in infectious diseases is complex and extends beyond simple inadequate ADH secretion [50].

### 4.2. Other—Dysnatremia in Critically Ill Patients

One of the included articles analyzed dysnatremia among patients admitted to the ICU [80]. This multi-center prospective study, which enrolled almost 14,000 patients (51% presented with infection on admission) has shown that 12.9% of patients presented low serum sodium concentration at the moment of inclusion. All kinds of hyponatremia, mild, moderate, and severe, were connected with higher mortality, irrespective of the presence or lack of infection. A positive relationship has been noticed between the severity of hyponatremia and in-hospital mortality. 

### 4.3. Drug—Induced Hyponatremia

Trimethoprim-sulfamethoxazole (TMP-SMX) is an antibiotic used mainly for the treatment of urinary tract infections and in the prevention of pneumocystis pneumonia. In a retrospective study, 72.3% of patients treated with this drug developed hyponatremia, 43.6% of which was moderate and severe [81]. After the termination of the therapy, serum sodium normalized within about 4 weeks in 50% of patients who had severe to moderate hyponatremia and in 60% of patients who had mild hyponatremia. Moreover, longer duration of therapy with TMP-SMX and its higher cumulative dose significantly lowered the serum sodium. Unfortunately, the pathomechanism of this state during TMP-SMX treatment remains unknown, though the authors suggest the main reason being renal salt wasting. An alternative mechanism has been suggested, that trimethoprim blocks sodium channels in the distal nephron, inducing excessive natriuresis [82]. 

## 5. Conclusions

There is no doubt that hyponatremia in infections is a multidisciplinary problem, as it may be associated with the infection itself, but also with a variety of factors such as renal or intracranial pathologies, drugs intake, malnutrition, or critical state. The knowledge about factors predisposing to hyponatremia remains essential for a proper diagnosis and treatment.

Many publications show that hyponatremia is associated with prolonged hospital stay and severity, as well as mortality in a number of infectious diseases. Despite many attempts to state its etiology in infective settings, few studies, just one in our review, took into consideration measuring the level of CT-ProVassopressin. This seems to be an adequate method to differentiate at least whether the low level of sodium is caused by SIADH or not. Therefore, we suggest including this test in the future studies of the pathomechanism of hyponatremia.

Several authors suggest that the correlation between hyponatremia during infection and mortality risk is causal, which might indeed be the case. However, it needs to be stressed that most of the infection-related deaths occur due to respiratory or circulatory failure, and not hyponatermia-associated neurologic damage. Thus, to prove the causal relationship between hyponatremia and mortality in infective settings, large prospective trials will need to be conducted in the future. Until then, in patients with infections hyponatremia should be interpreted as an indirect marker of disease severity, a potential direct marker of other concomitant disease, or a predictor of poor prognosis in this setting.

As mentioned in the introduction, it has been recently proven that improvement of hyponatremia reduces the risk of death among patients admitted to hospital due to this disorder. Therefore, there are no doubts that each case of hyponatremia should be treated causally in order to reinstate correct water–electrolyte balance in organism. Mild- and moderate hyponatremia in certain settings may be well tolerated and treated ambulatory. However, most of the studies indicate that patients with serum sodium threshold of <125 mmol/L (irrespective of their symptoms) should be immediately admitted to hospital and treated intensively, due to high mortality reaching up to 20% in these patients.

The role of hyponatremia in HIV-infection remains uncertain. These patients were mostly excluded from studies focusing on the impact of hyponatremia on prognosis and mortality, and so the results of these cannot be extrapolated on the population of HIV positive patients.

The novelties in our paper, comparing to the similar review released in 2011, include description of hyponatremia in Ebola-infected patients, in COVID-19 patients, and in influenza-infected patients. 

## Figures and Tables

**Figure 1 ijerph-17-05320-f001:**
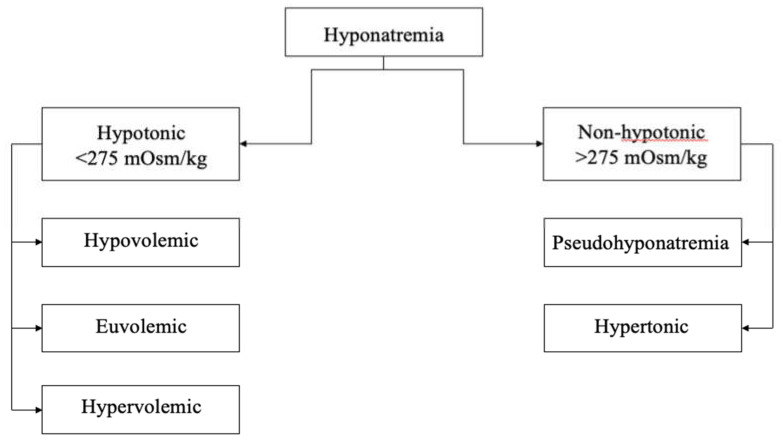
Types of hyponatremia.

**Figure 2 ijerph-17-05320-f002:**
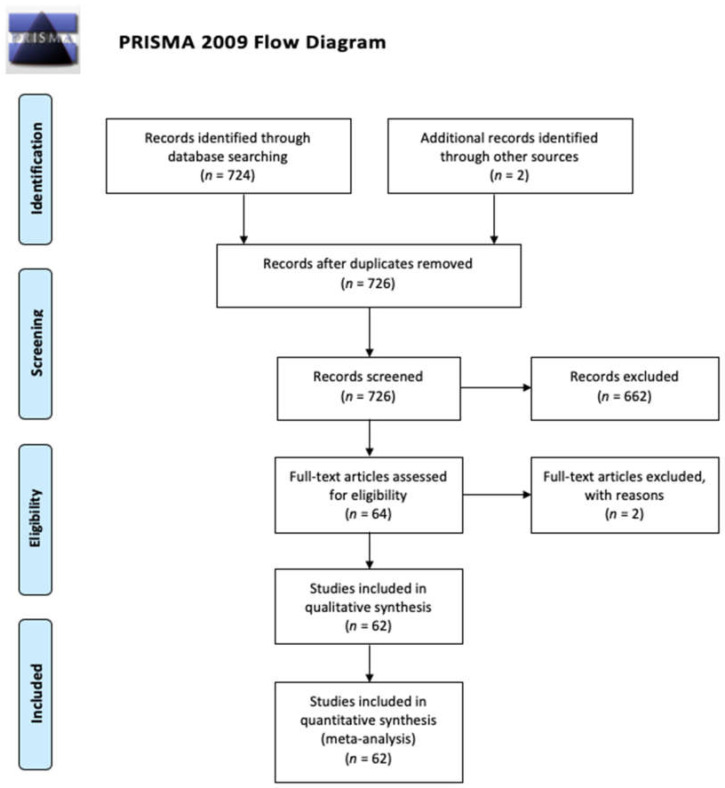
Preferred Reporting Items for Systematic Reviews and Meta-Analyses (PRISMA) 2009 flow diagram.

**Table 1 ijerph-17-05320-t001:** Summary of the most common infectious diseases occurring with hyponatremia.

Bacterial	Viral	Fungal	Protozoal
CAP—mainly caused by *Legionella pneumophila* SBP Tuberculosis Murine typhus Sepsis Meningitidis Skin and soft tissue infection	HIV/AIDS HHV-6 Hantavirus SFTSV Ebola virus CNS infections	Cryptococcosis	Malaria

Abbreviations: CAP—community acquired pneumonia; SBP—spontaneous bacterial peritonitis; HIV—human immunodeficiency virus; AIDS—acquired immune deficiency syndrome; HHV-6—human herpesvirus 6; SFTSV—severe fever with thrombocytopenia syndrome virus; CNS—central nervous system.

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
