# Peer review of "Hyponatremia in Infectious Diseases—A Literature Review"

_ijerph, 2020, doi:10.3390/ijerph17155320_

Round 1

Reviewer 1 Report

The article title is misleading as this is not a systematic review at all, but rather a narrative review. A systematic review should conform to all the requirements in PRISMA, not just the flow chart. There are no inclusion or exclusion criteria for articles or any defined process for data extraction or quality assessment of articles.

The stated aims are too vague, especially the aim to 'summarise the possible role of hyponatremia in the diagnosis of different disorders'. A systematic review of the prevalence of hyponatremia in infectious diseases (or a selection of infectious diseases) should have been deliverable, but that is not what has been presented.

The article lacks novelty and is very similar to the narrative review by Liamis et al from 2011 (ref 4). It is unclear how any further useful insights are added with this update. 

There is a disappointing lack of detail in the write-up on the prevalence of hyponatremia in each infection. Much of the write-up is a general overview of hyponatremia, and is not sufficiently focussed on infection.  

Author Response

Dear Reviewer,

Thank you for your valuable comments on the manuscript. All of your suggestions were taken into consideration. According to your recommendations, several changes have been applied:

  1. “The article title is misleading as this is not a systematic review at all, but rather a narrative review. A systematic review should conform to all the requirements in PRISMA, not just the flow chart. There are no inclusion or exclusion criteria for articles or any defined process for data extraction or quality assessment of articles.”

You are right – our article is not a systematic review indeed. However, it addresses a common and important clinical problem and we are convinced that it may be useful to certain physicians.

The primary purpose of this paper was to perform a meta-analysis. However, after selecting eligible papers and analyzing the available data, they were found insufficient for a meta-analysis our statistician. Although the paper does not strictly meet the definition of a systemic review, we have decided to still adhere to the PRISMA model to ensure the highest quality of the selected material and scientific content. This information is in the Introduction (line 54-60).

We have also mentioned the inclusion and exclusion criteria in the Materials and Methods section as you have suggested (line 69-75). Please, see the attachment as well – the PRISMA checklist, where all the details were described.

  1. “The stated aims are too vague, especially the aim to 'summarize the possible role of hyponatremia in the diagnosis of different disorders'. A systematic review of the prevalence of hyponatremia in infectious diseases (or a selection of infectious diseases) should have been deliverable, but that is not what has been presented.” –

Our review aims to summarize the possible role of hyponatremia and its prevalence in various infectious diseases. It is indeed a broad topic, however, we are convinced that many clinical practitioners will find it relevant, as the article addresses common medical problems. These include patients with community-acquired -pneumonia (overall incidence 68-7000 per 100,000) or patients with HIV-disease hospitalized due to concomitant infection (15%-per-patient year hospitalization risk) [1], [2].

Since the review focuses on several clinical problems associated with hyponatremia, no universal conclusion, that could be extrapolated on general patient-community could be drawn. However, the literature shows clearly that severe hyponatremia (<125 mmol/L) in this setting is usually a marker of disease severity and is associated with adverse outcome – “most of the studies indicate that patients with serum sodium threshold of <125mmol/L (irrespective of their symptoms) should be immediately admitted to hospital and treated intensively, due to high mortality reaching up to 20% in these patients”

  1. “The article lacks novelty and is very similar to the narrative review by Liamis et al from 2011. It is unclear how any further useful insights are added with this update.” –

We believe that, although the topic of hyponatremia is not novel, we have addressed several new articles and clinical settings where hyponatremia may occur. These novelties in our paper, compared to the cited paper, include the description of hyponatremia in Ebola-infected patients, in COVID-19 patients, and Influenza-infected patients. We have also discussed the role of measuring the level of CT-ProVassopressin, differentiating between Legionella pneumophila from Streptococcus pneumoniae or other non-legionella pneumophila infections.

Thus, we believe, that this review promotes novel findings that will be found relevant by clinicists worldwide and might stimulate further research in this field.

  1. “There is a disappointing lack of detail in the write-up on the prevalence of hyponatremia in each infection. Much of the write-up is a general overview of hyponatremia and is not sufficiently focused on infection.” –

You are right – the article lacked details on the prevalence of hyponatremia in each section. These data were supplemented in the review - please see lines 241-242, 305-308, 331-332, 409-410, 443-444 for more information.

We truly hope that now, after implementing changes in text in response to your revision, you will find the manuscript suitable for publication.

Best regards,

Adrianna Kruczkowska

On behalf of all authors.

  1. Torres A, Cillóniz C, Blasi F, et al. Burden of pneumococcal community-acquired pneumonia in adults across Europe: A literature review. Respir Med. 2018;137:6-13. doi:10.1016/j.rmed.2018.02.007
  2. Falster K, Wand H, Donovan B, et al. Hospitalizations in a cohort of HIV patients in Australia, 1999-2007. AIDS. 2010;24(9):1329-1339. doi:10.1097/QAD.0b013e328339e245

Reviewer 2 Report

This manuscript is, " Hyponatremia in infectious diseases - a systematic review " by Dr. Anna L. Królicka and his colleagues. This study is a review article dealing with numerous infection-causing microorganisms and hyponatremia. This issue is useful for clinician and so interested. The manuscript is well-written but the loose structure. In my opinion, there were several problematic concerns as the following.

The reconstruction for all the manuscript is necessary. The infectious disease is the aimed etiology as presented in your title, so the infectious causes of hyponatremia is only necessary for demonstration. Numerous paragraphs in the following sections should be detected: "5. SIADH" (Page 9) and "7. Drug-induced hyponatremia" (Page 10). In addition, several paragraphs discussed with the essential concept of hyponatremia, in terms of " 3. Pathogenesis of hyponatremia" (Page 2-10), should be switched to the section of "1. Introduction"; numerous descriptions (line 150-165) in the section of "4.1.1. Influenza virus" might be condensed only to highlight the hyponatremia-related pathogens in CAP; and paragraphs in the section of "8. Coronavirus" should be moved to the category of "4.1. Viral infections".

Some specific comments below: 
Page 1: numerous sentences should be reworded for condense or clarification, such as Line 11-13 and Line 41-42.

Page 6: Line 205-206. The reference about this sentence should be inserted.

Author Response

Dear Reviewer,

Thank you for agreeing to review our manuscript and for your relevant comments on the manuscript. We took into consideration all of your suggestions. According to your recommendations, several changes have been applied:

  • “The reconstruction for all the manuscript is necessary” – as you suggested, the manuscript was thoroughly restructured.

  • “several paragraphs discussed with the essential concept of hyponatremia, in terms of " 3. Pathogenesis of hyponatremia" (Page 2-10), should be switched to the section of "1. Introduction" – the changes were applied.

  • “numerous descriptions (line 150-165) in the section of "4.1.1. Influenza virus" might be condensed only to highlight the hyponatremia-related pathogens in CAP”- The section was condensed as much as possible, however relevant information was retained.

  • “paragraphs in the section of "8. Coronavirus" should be moved to the category of "4.1. Viral infections"” – Paragraph about Coronavirus was moved to the viral section. The latest reports about COVID-19 were also included in this section, as suggested by other reviewers.

  • “Page 1: numerous sentences should be reworded for condense or clarification, such as Line 11-13 and Line 41-42” – The whole abstract had been rewritten for better transparency and clarification of the papers aims and content.

  • “Page 6: Line 205-206. The reference about this sentence should be inserted” – The changes were applied.

Now that the suggested changes have been applied, we await more feedback from you about the paper and its suitability for publication.

Best regards,

Adrianna Kruczkowska

On behalf of all authors.

Reviewer 3 Report

The authors have made an updated review of a clinical frequent problem.

Introduction.  It would be of interest to address a summary of the epidemiology of hyponatremia: presentation percent in the hospital setting and in intensive care unit, were really it can be a challenge. Also, is there is a serum sodium threshold for morbidity and mortality?  Winzeler et al. (Eur J Endocrinol 2016) hast report that serum sodium values  reduction <125 mmol/L are associated with increased mortality 

The pathogenesis of hyponatremia is well described, and the classification could be more clearly related:3.3Hypotonic Euvolemic hyponatremia;3.3.1 SIADH.....

Hyponatremia due to infections.  The different causes are well described, but again, it would be helpful to have data on the percent of hyponatremia due to these agents in the clinical setting. Authors (page 5, second paragraph, that Hyponatremia was independently associated with disease severity. If possible remark the value of this association....In the third paragraph, : It may suggest, that serum sodium concentrations can be used as an indicator of the progression of HIV,,,, explain this fact.

The authors include Thyroid insufficiency and adrenal insufficiency, without relating the most frequent etiology in these conditions to cause hyponatremia. 

The concluding remarks,  are clear but can summarize, excluding critics as "Moreover, we notice a discrepancy between are articles..... It better to argued that low hyponatremia has been considered as a factor increasing mortality, while others consider hyponatremia a marker of the illness.

Author Response

Dear Reviewer,

Thank you for agreeing to review our manuscript and for your relevant comments on the manuscript. We took into consideration all of your suggestions. According to your recommendations, several changes have been applied:

  • “It would be of interest to address a summary of the epidemiology of hyponatremia: presentation percent in the hospital setting and in intensive care unit, were really it can be a challenge. Also, is there is a serum sodium threshold for morbidity and mortality?” – as you suggested, a summary of the epidemiology of hyponatremia has been included in the Introduction.

  • “The pathogenesis of hyponatremia is well described, and the classification could be more clearly related:3.3Hypotonic Euvolemic hyponatremia;3.3.1 SIADH” – The whole section about the pathogenesis of hyponatremia has been restructured and was moved to „Introduction” section, for more clarity for the potential reader.

  • “The different causes are well described, but again, it would be helpful to have data on the percent of hyponatremia due to these agents in the clinical setting.” – According to your suggestion, we added information about the prevalence of hyponatremia in every infection described in our manuscript.

  • “Authors (page 5, second paragraph, that Hyponatremia was independently associated with disease severity. If possible remark the value of this association....In the third paragraph, : It may suggest, that serum sodium concentrations can be used as an indicator of the progression of HIV,,,, explain this fact.” – We apologize for not making it clear enough. We reworked this paragraph to ensure that this subject is described explicitly now.

  • “The authors include Thyroid insufficiency and adrenal insufficiency, without relating the most frequent etiology in these conditions to cause hyponatremia.” – as these issues were not strictly associated with hyponatremia, and other reviewers suggested to reduce the description of pathogenesis, we have decided to delete this part. Brief information about these potential causes of hyponatremia were included in appropriate sections.

  • “The concluding remarks, are clear but can summarize, excluding critics as "Moreover, we notice a discrepancy between are articles..... It better to argued that low hyponatremia has been considered as a factor increasing mortality, while others consider hyponatremia a marker of the illness.” – after your suggestion, we decided to give up on criticism, which was unintentional. The remarks have been restructured following your suggestions.

We believe, that now, after the changes that were applied, you might find the manuscript suitable for publication.

Best regards,

Adrianna Kruczkowska

On behalf of all authors.

Reviewer 4 Report

This is an interesting review. However:

  • Pathogenesis of hyponatremia is pointlessly long: I suggest to reduce it.
  • All bacteria must be written in italic.
  • I would suggest to follow this order: viral infections, fungal infections, bacterial infections and protozoal infections
  • In consideration of the epidemiology of COVID-19, I strongly suggest to read and add the following articles:
  • Y Wu Front Med 2020 9 7 315
  • F Ata BMJ Case Report 2020 7 13
  • M Christ-Crain Eur J Endocrinol 2020 183 G9
  • A Berni J Endocrinol Invest 2020 25 1
  • E Tantisattamo Transpl Infect Dis 2020 8 e13355
  • English tongue should be revised here and there 

Author Response

Dear Reviewer,

Thank you for agreeing to review our manuscript and for your relevant comments on the manuscript. We took into consideration all of your suggestions. According to your recommendations, several changes have been applied:

  • “Pathogenesis of hyponatremia is pointlessly long: I suggest to reduce it” – the whole section has been modified and restructured, to make it brief and clear.

  • “All bacteria must be written in italic” - The error has been corrected.

  • “I would suggest to follow this order: viral infections, fungal infections, bacterial infections, and protozoal infections” – the following order was applied, and the „COVID-19” section was included within „viral infection section”

  • “In consideration of the epidemiology of COVID-19, I strongly suggest to read and add the following articles:

Y Wu Front Med 2020 9 7 315

F Ata BMJ Case Report 2020 7 13

M Christ-Crain Eur J Endocrinol 2020 183 G9

A Berni J Endocrinol Invest 2020 25 1

E Tantisattamo Transpl Infect Dis 2020 8 e13355” – The articles are interesting indeed and bring some insight into the epidemiology of COVID-19. However, only two of them: Y Wu Front Med 2020 9 7 315, and A Berni J Endocrinol Invest 2020 25 1  could have been included in our paper, following the previously applied inclusion and exclusion criteria. For more information please see the Materials and Methods section, line 185-191.

  • The manuscript has been corrected and restructured and finally sent for internal language correction within our institution, as to ensure high quality of the written language.

We believe, that now after the changes were applied, you might find the manuscript suitable for publication.

Best regards,

Adrianna Kruczkowska

On behalf of all authors.

Round 2

Reviewer 1 Report

I appreciate the authors have substantially rewritten their manuscript in light of my suggestions but I remain of the view there is insufficient novelty and scientific rigour for an original publication (as opposed to a commissioned expert review on the topic). Whether the manuscript is of sufficient clinical interest to justify publication in this journal is an editorial decision. 

This manuscript is a resubmission of an earlier submission. The following is a list of the peer review reports and author responses from that submission.